# Cycle Characteristics and Pregnancy Outcomes of Early Rescue Intracytoplasmic Sperm Injection Cycles in Normal and Hyper-Ovarian Response Women: A Six-Year Retrospective Study

**DOI:** 10.3390/jcm12051993

**Published:** 2023-03-02

**Authors:** Liang Chen, Hanjing Zhou, Xueli Liu, Jing Zhao, Qianrong Qi, Qingzhen Xie

**Affiliations:** 1Center for Reproductive Medicine, Renmin Hospital of Wuhan University, Wuhan 430060, China; 2Urology and Andrology Department, Renmin Hospital of Wuhan University, Wuhan 430060, China

**Keywords:** in vitro fertilization, fertilization failure, rescue intracytoplasmic sperm injection, live birth

## Abstract

This study aims to analyze the cycle characteristics, pregnancy, and neonatal outcomes in early rescue intracytoplasmic sperm injection (r-ICSI) cycles in normal and hyper-ovarian response women in their first IVF/ICSI attempts. Data from short-term in vitro fertilization (IVF, N = 7148), early r-ICSI (N = 618), and ICSI (N = 1744) cycles were retrospectively analyzed from normal and hyper-ovarian women who underwent their first IVF/ICSI cycles at our center from October 2015 to October 2021. The r-ICSI group was subdivided into partial r-ICSI (N = 451) and total r-ICSI (N = 167) based on the number of fertilized oocytes in the IVF part. Cyclic characteristics, pregnancy, delivery and neonatal outcomes in the fresh cycle were compared among the four groups; pregnancy, delivery and neonatal outcomes in frozen-thawed cycles were compared regarding cleavage and blastocyst transfers derived from r-ICSI cycles. Partial r-ICSI cycles showed different cyclic characteristics compared to total r-ICSI cycles, presenting as elevated AMH and estradiol levels on trigger day and an increased number of oocytes retrieved. Early r-ICSI delayed blastocyst development as seen by the increase in the number of day 6 blastocysts. There was no significant difference among the groups in clinical pregnancy, pregnancy loss, and live birth in fresh cleavage-stage embryo transfer cycles. However, early r-ICSI groups showed a reduction in clinical pregnancy and live birth rates in fresh blastocyst transfer cycles but not in the frozen-thawed cycles. For pregnant women, early r-ICSI did not show a negative effect on the risk of preterm birth, Cesarean section, neonatal birth weight, and sex ratio. In conclusion, early r-ICSI had comparable pregnancy, delivery, and neonatal outcomes when compared with short-term IVF and ICSI groups in fresh cleavage-stage embryo transfer cycles, but early r-ICSI did result in reduced pregnancy outcomes in fresh blastocyst embryo cycles, possibly due to delayed blastocyst development and asynchronization with the endometrium.

## 1. Introduction

Fertilization is a complex physiological process that successively includes sperm penetration, extrusion of the second polar body (PB), oocyte activation, decondensation of maternal and paternal nuclei, and cytoplasmic chromosome migration of the pronuclear material [1]. A fault with any of the individual steps could cause fertilization failure, thus leading to infertility at the very beginning step [2]. Despite the success of assisted reproductive technology (ART) for infertile couples, fertilization failure is the most frustrating situation for embryologists and leads to cycle cancellation, economic loss, and much distress for the patients. The incidence of total fertilization failure (TFF) after in vitro fertilization (IVF), in cases with normal sperm quality, has been reported to range from 5% to as high as 15–20% [3]. Intracytoplasmic sperm injection (ICSI) was introduced in the early 1990s to treat severe male infertility by injecting sperm into the oocytes [4], with a significantly reduced incidence of TFF [5]. ICSI application in ART cycles has increased rates to close to 80% in the United States and approximately 100% in the Middle East [6]. The reliability in fertilization represents a plausible reason for the expansion of ICSI over IVF for non-male infertility indications. However, excessive employment of ICSI in couples without male factor infertility, aiming at preventing TFF, is still debated and thought to have non-superior impacts on pregnancy outcomes, with higher costs and increased workload [7,8]. 

Fertilization failure after IVF is proposed to be due to the failure of a sperm to penetrate into the oocytes. Rescue ICSI (r-ICSI) was first introduced in 1993 for insemination on the second day after IVF failure, solving sperm penetration issues. However, late r-ICSI on 1-day-old oocytes yields poor fertilization and pregnancy outcomes, probably due to oocyte aging and asynchronization between embryo development and endometrial receptivity. Late r-ICSI is associated with poor pregnancy outcomes in the fresh cycle [9], while embryos derived from late r-ICSI are suggested to be transferred in a frozen-thawed cycle with improved implantation rates and clinical pregnancy rates [10,11]. Based on the published data, it is necessary to notice that the overall success rate of late r-ICSI is in the range of “futility” or “very poor prognosis” [12]. 

Short-term IVF combined with early r-ICSI at 4–6 h after insemination could improve the fertilization outcomes and avoid oocyte aging since the second PB is extruded in 80% and 90% of fertilized oocytes by 4 and 6 h, respectively [13]. In addition, to prevent TFF, short-term IVF with early r-ICSI can shorten the incubation time of oocytes in a potential detrimental environment from metabolic sperm degradation products. In clinical practice, however, given the increased burden of laboratory workflow implementation, the early r-ICSI technique is currently quite unpopular. Hitherto, limited studies focused on the pregnancy outcomes from early r-ICSI, the lack of rigorous inclusion and exclusion criteria, clinical information, and obstetric and neonatal outcomes, hindering the application of early r-ICSI. Given this scenario, it is worth depicting the beneficial roles of early r-ICSI in order to assist embryologists in proceeding with IVF in non-male factor infertility. Our center has been performing short-term IVF and early r-ICSI for non-male factor infertile couples in their first IVF cycle for over ten years. The purpose of this study was to analyze the cyclic characteristics between short-term IVF, r-ICSI, and ICSI cycles from normal and hyper-ovarian response women in their first IVF attempts and identify the potential risk factors of fertilization failure. The pregnancy, delivery, and neonatal outcomes were compared among the groups to evaluate the safety and efficacy of r-ICSI on pregnancy outcomes.

## 2. Materials and Methods

### 2.1. Study Design

Figure 1 presents the patient recruitment flow in this study. Data from 15,307 IVF/ICSI cycles in the Reproductive Medicine Center, Renmin Hospital of Wuhan University, between October 2015 and October 2021 were retrospectively screened. The first IVF/ICSI attempts of 10,151 women with normal and hyper-ovarian responses (anti-Mullerian hormone (AMH) ≥ 1.1 ng/mL, number of oocytes retrieved ≥ 5, age ≤ 42) were analyzed. Among them, 939 cycles from a poor ovarian response and/or decreased ovarian reserve women (AMH < 1.1 ng/mL, number of oocytes retrieved < 5, and age > 42), 241 cycles for testicular sperm aspiration (TESA)/ICSI, 354 cycles for preimplantation genetic testing (PGT)/ICSI, 18 cycles for half IVF/ICSI, and 28 cycles for oocyte cryopreservation were excluded. A total of 7148 cycles of short-term IVF, 618 cycles of r-ICSI, and 1744 cycles of ICSI were included for final analysis. Based on the fertilization results in the IVF group, the r-ICSI group was subdivided into a partial r-ICSI group (at least one fertilized oocyte in IVF, N = 451) and a total r-ICSI group (presented as TFF after IVF, N = 167). A total of 572 frozen-thawed embryo transfers derived from early r-ICSI cycles between October 2015 and October 2021 were retrospectively analyzed. 

### 2.2. Controlled Ovarian Stimulation and Oocyte Retrieval

Infertile couples were required to establish an electronic record after a comprehensive evaluation of their infertility reasons and pre-pregnancy examinations. In this study, the infertility reasons were divided into female infertility (including fallopian tube factors, ovulation disorders, endometriosis, and other confirmable female infertility reasons), male infertility (abnormal sperm quantity or quality, donor semen due to azoospermia, and sexual dysfunction), and unexplained infertility (no confirmative infertility reason was identified after comprehensive evaluation). 

The patients underwent ovarian stimulation with either a follicular phase gonadotropin-releasing hormone agonist (GnRHa) protocol, luteal phase GnRHa protocol, GnRH antagonist protocol, or a progestin-primed ovarian stimulation protocol (PPOS) using recombinant FSH stimulation according to the guidelines in our center. FSH doses were adjusted individually to the patients’ ovarian responses. A dose of 8000 IU to 10,000 IU of hCG was administered at 8:00–11:00 p.m. when at least three follicles measured ≥17 mm, and ultrasound-guided oocyte retrieval was conducted 36 h later (08:00–11:00 a.m.). All patients signed an informed consent for IVF/ICSI treatment and follow-up. 

### 2.3. Sperm Preparation 

All the male partners were required to undergo semen analysis twice two at the clinic to evaluate their insemination options. The ICSI technique was reserved if semen parameters met any of the following conditions: (1) spermatozoa concentration < 1.5 × 10^6^/mL; (2) percentage of progressive motile (PR) spermatozoa < 25% or total motile spermatozoa < 40% when spermatozoa concentration was <20 × 10^6^/mL; (3) percentage of total motile spermatozoa < 5% when spermatozoa concentration was >20 × 10^6^/mL; (4) severe teratospermia (normal spermatozoa < 1%). Short-term IVF was the primary insemination option for other semen conditions. 

Semen samples were collected by masturbation after 3–5 days of sexual abstinence at the time of oocyte retrieval and liquefied for at least 30 min at 37 °C. Liquefied semen were analyzed by spermatozoa count plate under a light microscope to record volume, spermatozoa concentration, and percentage of PR spermatozoa, then prepared by conventional discontinuous density gradient centrifugation and swim-up procedures in accordance with the manufacturer’s instructions. 

### 2.4. ICSI Procedure

Cumulus–oocyte complexes (COCs) were cultured in equilibrated fertilization medium (COOK, Sydney, Australia) for 2–4 h following retrieval, then processed with hyaluronidase (Vitrolife, Gothenburg, Sweden) at 37 °C for 30 to 60 s to remove the cumulus. The oocytes were then transferred to the fertilization medium and denuded by a ready-to-use pipette. Centrifuged spermatozoa were added into the ICSI operational medium and processed with sperm injection into matured oocytes (MII oocytes) at 13:00–16:00 pm. Briefly, the MII oocyte was immobilized by holding pipettes (Sunlight Medical, Jacksonville, FL, USA) with the first PB at 12 o’clock, and the immobilized sperm was microinjected into the cytoplasm at 3 o’clock by ICSI injection pipettes (Sunlight Medical, USA). The inseminated oocytes were then transferred into a pre-balanced COOK embryo culture system (COOK) for embryo culture according to the manufacturer’s instructions. 

### 2.5. Short-Term IVF Procedure, Early Second PB Assessment, and Early r-ICSI

Cumulus–oocyte complexes (COCs) were cultured in equilibrated fertilization medium (COOK) for 2–4 h following retrieval and then inseminated with motile spermatozoa at the concentration of 10,000 spermatozoa in 40–50 µL fertilization medium droplets covered by mineral oil (COOK, Sydney, Australia) at the embryo incubator (COOK). Specifically, the oocytes were retrieved at 08:00–11:00 am and co-cultured with spermatozoa at 12:00–13:00. The second PB (2PB) assessment was performed at 16:00–18:00 in our center. Cumulus cells were mechanically removed 4–6 h after insemination by a ready-to-use pipette (COOK) for early PB assessment under an inverted microscope (Leica, 400×). The oocytes with 2PB were confirmed as successfully fertilized, and early r-ICSI was performed if the rate of oocytes with 2PB was less than 30% in all MII oocytes. Motile sperm with normal morphology were microinjected into unfertilized MII oocyte cytoplasm as mentioned above for conventional ICSI. The inseminated oocytes were then transferred into a pre-balanced COOK embryo culture system (COOK) for embryo culture according to the manufacturer’s instructions. 

### 2.6. Fertilization Assessment, Embryo Quality Categorization, and Embryo Transfer

On the first day (07:00–09:00), pronuclear morphology was observed, including the number of pronuclei (PN) and polarization of nucleolus precursor bodies. The morphology and development of all embryos were evaluated and graded on the morning of the third day, fifth day, and sixth day. A top-quality embryo on day 3 was defined as an embryo developed from 2PN with seven or eight blastomeres with less than 20% fragmentation; blastocysts were graded based on the expansion of the blastocoel cavity, the number of cells, and the cohesiveness of the inner cell mass and trophectoderm according to Gardner criteria [14]. Top-quality blastocysts were embryos were graded as AA, AB, BA, or BB. One or two top-quality embryos were transferred on day 3 or day 5, and the surplus embryos were vitrified for cryopreservation. 

### 2.7. Endometrial Preparation Protocol for Frozen-Thawed Embryo Transfers

The endometrium preparation protocols for frozen-thawed embryo transfers (FET) included the natural cycle, hormone replacement treatment (HRT) cycle, or GnRHa + HRT cycle. In the natural cycle, endometrium preparation was achieved by follicle growth and LH monitoring [15]. HRT cycles for endometrium preparation consisted of treatment with estradiol valerate (ProgynovaVR; Bayer-Schering Pharma AG, Berlin, Germany) 2 mg twice daily for 7 days, followed by 2 mg three times daily for 6 days. Progesterone supplementation was started on day 13 if the endometrium was at least 7 mm thick, a triple-line endometrium was present, and serum progesterone levels were <1.5 ng/mL. Day 3 embryos were transferred on the fourth day of progesterone exposure, and the blastocysts were transferred on the sixth day of progesterone exposure. For GnRHa + HRT cycles, 3.75 mg leuprorelin acetate (DipherelineVR, Ipsen, France) or 3.75 mg triptorelin acetate (DecapeptylVR, Ferring, Switzerland) was administered during the early follicular phase of the previous menstrual cycle (day one or two), and the HRT protocol was started 28 days later [16]. 

### 2.8. Assessment of Pregnancy Outcomes

Serum beta-hCG levels were measured between 10 and 14 days following embryo transfer. If the test was positive, daily estradiol valerate and progesterone supplementation were continued until the 12th week of pregnancy. An ultrasound scan was carried out 30–35 days after embryo transfer to determine fetal viability. Clinical pregnancy was defined as the presence of at least one fetus with a heartbeat on ultrasound 30–35 days after embryo transfer. The pregnancy outcomes were collected through clinic visits and telephone follow-ups, including spontaneous abortion, ectopic pregnancy, delivery conditions, and neonatal status.

### 2.9. Ethical Approval 

The present study retrospectively analyzed the electronic and paper database in our hospital. All the participants (wife and husband) signed informed consents regarding controlled ovarian hyperstimulation, oocyte and semen collection, IVF or ICSI treatment, embryo cryopreservation, embryo transfer, and follow-up visit management. All the procedures complied with the *Regulation of Human-Assisted Reproductive Technology in China*. Institutional review board approval was obtained from Wuhan University Renmin Hospital (WDRY2022-K265).

### 2.10. Statistical Analysis

All the information from the baseline characteristics, ovarian stimulation, oocyte retrieval, laboratory procedure and assessment, embryo transfer, and follow-up data were recorded in an electronic system in detail. Data were presented as mean ± SD for measurement data and case number and percentage for enumeration data. Statistical analysis was performed using SPSS version 19.0 (IBM). Quantitative variables were analyzed by Student’s *t*-test and one-way analysis of variance (ANOVA) to compare the differences, and the least square difference test was used for post hoc comparisons. Categorical variables were analyzed by Pearson’s chi-squared test or Fisher’s exact test. *p* < 0.05 indicated statistical significance. 

## 3. Results

### 3.1. The Baseline and Cycle Characteristics in the Different Insemination Groups

The present study included 9510 non-poor ovarian response patients who underwent their first IVF/ICSI cycle. Among them, 7766 infertile patients were assigned to short-term IVF with normal spermatozoa parameters on the day of oocyte retrieval, and 1744 patients received ICSI due to male factor infertility. After short-term IVF and early 2PB check, 451 patients received partial early r-ICSI because the rate of oocytes with 2PB was less than 30% in all MII oocytes, and 167 patients received total early r-ICSI due to TFF. The incidence of early r-ICSI after IVF was 7.96% (618/7766), and the incidence of TFF after IVF was 2.15% (167/7766).

The patients’ clinical characteristics of short-term IVF, partial r-ICSI, total r-ICSI, and ICSI groups are presented in Table 1. There was no significant difference in paternal age, BMI, COH protocols, total Gn dosage, Gn stimulation, progesterone, and LH levels on hCG trigger day among the four groups. The maternal age, AMH, and estradiol (E2) level on hCG trigger day were comparable among IVF, total r-ICSI, and ICSI groups. The partial r-ICSI group showed younger age and significantly elevated AMH and estradiol levels on hCG trigger day (*p* < 0.001). Partial and total r-ICSI groups showed remarkably prolonged infertility duration, an increased proportion of primary infertility, and unexplained infertility. 

### 3.2. Embryonic Laboratory Outcomes in the Different Insemination Groups

Table 2 focuses on the fertilization, embryo quality, and cyclic outcomes among the four groups. The average number of retrieved oocytes was significantly higher in the partial r-ICSI group (15.27 ± 6.54, *p* < 0.001) compared to the IVF (14.23 ± 7.26) and total r-ICSI groups (11.06 ± 6.87). The average fertilization rate of 2PN was comparable in the short-term IVF (62.03 ± 20.48) and total r-ICSI (65.31 ± 23.09) groups, and the polyspermy rate (≥ 3PN) was remarkably reduced in r-ICSI groups compared with the IVF group (*p* < 0.001). The IVF group demonstrated an increased number of top-quality embryos on D3 and a higher blastocyst formation rate compared to r-ICSI and ICSI groups. Notably, the number of day 5 blastocysts in r-ICSI groups was lower, but the number of day 6 blastocysts was increased in comparison with the IVF and ICSI groups, suggesting that postponed re-insemination by early r-ICSI may delay blastocyst development. The ICSI group showed a lower proportion of fresh embryo transfer cycles and increased freeze-all cycles compared to the IVF group. The TFF rate was comparable among the four groups, while total r-ICSI and ICSI groups showed increases in cycles without available embryos for transfers or cryopreservation.

### 3.3. The Pregnancy, Delivery, and Neonatal Outcomes in the Different Insemination Groups

Table 3 shows the pregnancy, delivery, and neonatal outcomes among the four groups in fresh cleavage-stage embryo transfer cycles. Compared to the IVF and ICSI groups, early r-ICSI did not exhibit any negative effect on the clinical pregnancy, early pregnancy loss, live birth (Figure 2A), and birth weight in fresh cleavage-stage embryo transfers, and early r-ICSI did not increase the risk of preterm birth and Caesarean section when compared with IVF and ICSI groups. Compared to the IVF group, r-ICSI and ICSI groups had a decreased ratio of male neonates (*p* = 0.002). In contrast to fresh cleavage-stage embryo transfers, early r-ICSI groups showed decreased clinical pregnancy, implantation, and live birth rates compared to IVF and ICSI groups in fresh blastocyst transfer cycles (*p* < 0.001, Table 4, Figure 2A). The ICSI group showed an increased preterm birth rate in comparison with the IVF group. No significant difference was found in the twin birth rate, birth weight, and sex ratio among the four groups. 

Since early r-ICSI groups showed delayed blastocyst formation and poor pregnancy outcomes in fresh blastocyst transfer cycles, we further analyzed the frozen-thawed embryo transfers (FET) derived from early r-ICSI cycles. A total of 572 FER cycles were included in Table 5. Compared to frozen-thawed cleavage-stage embryo transfers, blastocyst transfers had a significantly decreased average number of embryos transferred and an increased implantation rate (*p* < 0.001). In contrast to the inferior pregnancy and live birth rates in fresh blastocyst transfers obtained from early r-ICSI, frozen-thawed blastocyst transfers showed similar rates in clinical pregnancy and live births (Figure 2B). No significant difference was found in the risk of preterm birth and Caesarean section and low birth weight between cleavage-stage and blastocyst transfers.

## 4. Discussion

ICSI, one of the most remarkable technological breakthroughs in ART, was initially developed to conquer severe male infertility but is currently widely applied in infertile couples without an indication of male factor infertility due to the guaranteed fertilization rate [17]. An open-label, randomized controlled study showed that ICSI did not improve the live birth rate compared with conventional IVF in infertile couples without male factor infertility, which challenges the value of the routine use of ICSI in ART [18]. Lately, clinical evidence has supported the need to limit the extensive use of ICSI and stated that the use of ICSI should be reserved only for male factor infertility [18]. Among the strategies that encourage the use of conventional IVF, rescue ICSI, by performing ICSI after 4 to 24 h on unfertilized oocytes, has been proposed to effectively rescue cycles with TFF or near TFF [19]. Given the implementation difficulties, early r-ICSI is currently not the preferred option compared to late r-ICSI, even though late r-ICSI is associated with a low success rate and deterioration in oocyte quality and embryo development. In addition, the cyclic outcomes, clinical safety, and efficacy of early r-ICSI have been poorly investigated to date. 

A previous study showed a reduced total number of top-quality embryos yielded on day 3 from early r-ICSI in women with primary infertility compared with conventional ICSI (4.02 vs. 5.15). No significant difference was found in pregnancy and delivery outcomes between the two groups [20]. Another long-term retrospective study suggested similar clinical pregnant and neonatal outcomes with early r-ICSI compared with IVF and ICSI; however, the inclusion and exclusion criteria were not well-defined in this study, and no specific analysis was performed on blastocyst transfers and frozen embryo transfer cycles [21]. In this study, we performed a retrospective 6-year analysis on embryonic laboratory parameters as well as pregnancy and delivery outcomes from early r-ICSI cycles in non-poor ovarian women during their first IVF attempts. The main results showed that early r-ICSI has no negative effect on pregnancy, delivery, and neonatal outcomes in fresh cleavage-stage transfer cycles, which is similar to previously reported pregnancy outcomes [20,21]. Interestingly, the clinical pregnancy and live birth rates from fresh blastocyst transfers were dramatically reduced in r-ICSI groups compared with IVF and ICSI groups, even though the fresh blastocyst transfer showed superior pregnancy and delivery outcomes compared to cleavage-stage embryos in IVF and ICSI groups. The possible reasons for these inferior pregnancy outcomes could be delayed blastocyst development and the time shift in fecundation. The blastocyst formation rate and top-quality blastocyst rate were decreased in r-ICSI groups compared to the IVF group, indicating that delayed fertilization by 5–6 h may postpone embryo development at the blastocyst stage and possibly lead to asynchronization between the blastocyst and endometrial receptivity. However, the sample size for fresh blastocyst transfers in r-ICSI groups was small. The reasons for dissatisfying pregnancy outcomes in fresh blastocyst transfer and the underlying mechanisms need to be further verified.

ICSI or half ICSI has been proposed for use in couples with primary and unexplained infertility since the fertilization barrier could be the causative factor for primary unexplained infertility [22]. However, evidence of improved fertilization and pregnancy outcomes with the use of ICSI, rather than IVF, for primary and unexplained infertility is still debated [23]. In this study, although the proportion of patients with primary infertility in the r-ICSI groups was higher than in the IVF group, the actual number of cases (417 cases) was about one-tenth of the IVF group (3419 cases). Likewise, the actual quantity of couples with unexplained infertility (128 cases) was three-tenths of the IVF group (420 cases) despite the increased proportion. The normal fertilization (2PN) rates were acceptable both in the IVF and r-ICSI groups (62.03%, 57.75%, and 65.31%, respectively), although lower than the ICSI group (67.61%), while the embryo laboratory data and the overall pregnancy outcomes were similar among the IVF, r-ICSI, and ICSI groups, and TFF occurred in 21 couples (0.27%) in the IVF group after r-ICSI and 8 couples (0.45%) in the ICSI group. A prospective clinical study randomized 60 unexplained infertile couples with IVF or ICSI, and their results showed no significant differences in fertilization rate, embryo quality, implantation rate, and live birth rate [22]. A retrospective analysis of 1130 half-ICSI treatments found no improvements in pregnancy outcomes in ICSI-originated embryo transfers in spite of superior fertilization rates and embryo quality [22]. Currently, the evidence supporting the routine or preventive use of ICSI for primary and unexplained infertility is limited and does not reveal improvements in clinical outcomes. Early r-ICSI combined with IVF successfully prevents the occurrence of fertilization failure and does not affect embryo quality and clinical outcomes. 

Late r-ICSI, re-insemination on 1-day-old oocytes, has been proposed as a supplementary strategy to prevent cycle cancellation due to TFF, in spite of the fact that it may result in an increased proportion of polyspermy, poor embryo quality with genetic abnormalities, and poor pregnancy outcomes [24]. Except for the possibility of double insemination of oocytes that were already fertilized due to disappearing or delayed pronuclei formation, the 3PN formation also increased because of oocyte aging [25]. Reducing the time between oocyte retrieval and r-ICSI on aged oocytes can yield a higher rate of normal fertilization, as well as improved embryo quality and pregnancy outcomes. Traditionally, in our laboratory, the early fertilization sign of 2PB is checked at 4 p.m. (4 h later after insemination), and r-ICSI is performed on oocytes without 2PB at 5–6 p.m. if the proportion of oocytes with 2PB is less than 30%. Our data showed the rate of 3PN is significantly reduced in r-ICSI groups compared to the IVF group, and the 2PN rate was comparable to the IVF group. Therefore, checking the second PB at 4 to 6 h after insemination can precisely evaluate the fertilization failure, and early r-ICSI does not increase the risk of polyspermy. 

The limitations of this study include the retrospective nature of the r-ICSI procedure, as well as the fact that the incidence of r-ICSI is relatively low in IVF cycles, resulting in the small sample size in the r-ICSI groups compared to the IVF and ICSI groups, especially the small sample size in fresh blastocyst transfer cycles. Considering the controversies in strictly distinguishing normal and hyper-response women in clinical practice [26,27], the data from normal and hyper-ovarian responses were not analyzed separately. In addition, fertilization failure also occurred in repeated cycles, which has a normal fertilization rate by IVF insemination, and aging has been correlated with zona pellucida thickening, which is also a risk for fertilization failure, while women with repeated cycles and poor ovarian response were excluded from this study. Further studies are needed to explore the risk factors and cyclic characteristics for fertilization failure and the outcomes of r-ICSI in women with repeated cycles and poor ovarian response. 

## 5. Conclusions

In conclusion, we focused on the cyclic characteristics and pregnancy and neonatal outcomes in early r-ICSI from normal and hyper-ovarian response women during their first IVF attempts. Our finding showed that increased ovarian reserve function (young age, high AMH, and increased number of oocytes retrieved) may be a risk factor for partial FF but not for TFF. The pregnancy, delivery, and neonatal outcomes were comparable in fresh cleavage-stage embryo transfers from r-ICSI cycles compared with IVF and ICSI cycles. However, postponed insemination by r-ICSI may potentially delay blastocyst development, thus resulting in a reduction in the live birth rate in fresh blastocyst transfers. Therefore, blastocysts derived from r-ICSI cycles are suggested for frozen-thawed transfer cycles. Our study demonstrates that early r-ICSI is an efficacy and safe option for embryologists in proceeding with conventional IVF in couples without male infertility.

## Figures and Tables

**Figure 1 jcm-12-01993-f001:**
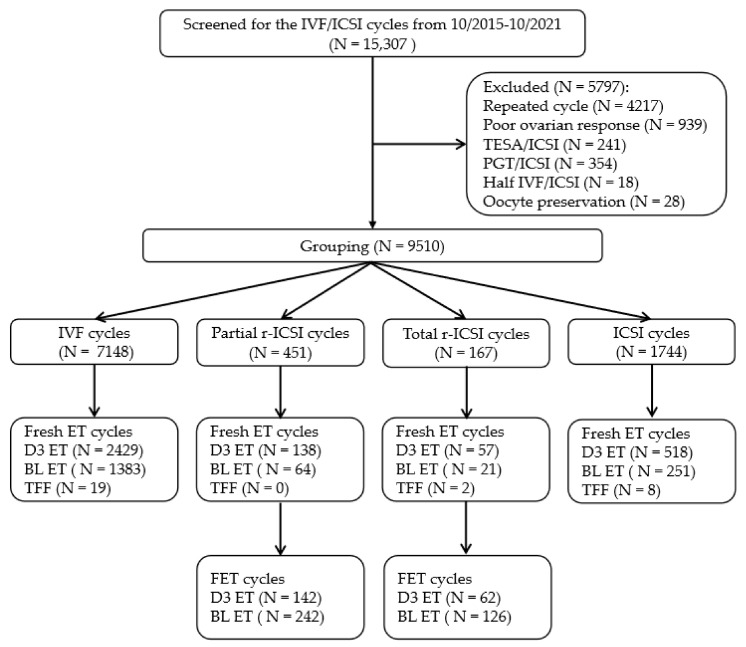
Cycles included in this study. IVF, in vitro fertilization; ICSI, intracytoplasmic sperm injection; TESA, testicular sperm aspiration; PGT, preimplantation genetic testing; ET, embryo transfer; FET, frozen-thawed embryo transfer; BL, blastocyst; TFF, total fertilization failure.

**Figure 2 jcm-12-01993-f002:**
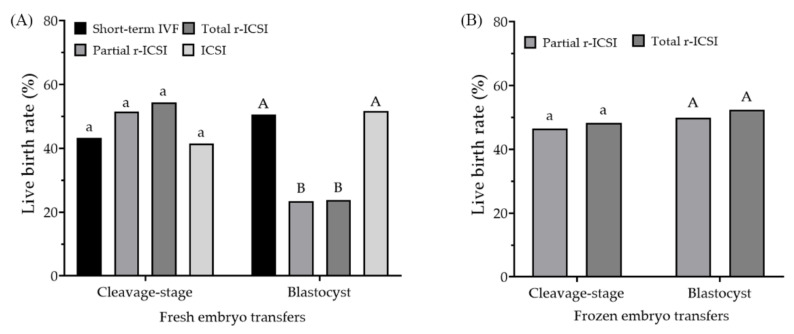
(**A**) Comparison of live birth rates per fresh embryo transfers from cleavage-stage and blastocyst transfers among the four groups. (**B**) Comparison of live birth rates per frozen embryo transfers from r-ICSI derived cleavage-stage and blastocyst transfers. Bars with different letters differ significantly among groups (*p* < 0.05).

**Table 1 jcm-12-01993-t001:** Baseline characteristics of couples among four groups.

	Short-Term IVF	Partial r-ICSI	Total r-ICSI	ICSI	*p*-Value
Cycles	7148	451	167	1744	
Maternal age (years)	31.61 ± 3.94 ^a^	30.9 ± 4.00 ^b^	31.92 ± 4.06 ^a^	31.45 ± 4.44 ^a^	<0.01
Paternal age (years)	33.68 ± 4.82 ^a^	33.74 ± 4.85 ^a^	33.88 ± 4.75 ^a^	34.49 ± 5.08 ^b^	<0.01
BMI (kg/m^2^)	22.19 ± 3.94	22.1 ± 4.01	22.37 ± 4.06	22.08 ± 4.44	NS
AMH (ng/mL)	3.89 ± 2.85 ^a^	4.55 ± 2.89 ^b^	4.10 ± 4.63 ^a^	3.78 ± 2.61 ^a^	<0.01
Infertility duration	3.24 ± 2.41 ^a^	3.78 ± 2.28 ^b^	4.28 ± 3.17 ^b^	3.44 ± 2.51 ^a^	<0.01
Infertility type, n (%)					<0.01
Primary infertility	3419 (47.83)	296 (65.63)	121(72.46)	999 (57.28)	
Secondary infertility	3729 (52.17)	155 (34.37)	46 (27.54)	745 (42.72)	
Infertility diagnosis, n (%)					<0.01
Tubal factor	4475 (62.60)	194 (43.02)	78 (46.71)	/	
Ovulation dysfunction	1302 (18.21)	107 (23.73)	30 (17.96)	/	
Endometriosis	823 (11.51)	45 (9.98)	11 (6.59)	/	
Male factor	128 (1.79)	21 (4.66)	4 (2.4)	1744	
Unexplained infertility	420 (5.88)	84 (18.58)	44 (26.35)	/	
COH protocols, n (%)					NS
Follicular-phase GnRHa	3064 (42.87)	195 (43.24)	69 (41.32)	731 (41.92)	
Luteal-phase GnRHa	1943 (27.18)	112 (24.83)	44 (26.35)	444 (25.46)	
GnRH antagonist	1991 (27.85)	123 (27.27)	42 (25.15)	460 (26.38)	
PPOS	420 (5.88)	21 (4.66)	12 (7.19)	109 (6.25)	
Total Gn dosage (IU)	2105 ± 757	2012 ± 710	2115 ± 724	2131 ± 717	NS
Gn stimulation days	10.64 ± 2.2	10.63 ± 2.25	10.67 ± 2.21	10.64 ± 2.19	NS
Estradiol on hCG day (pg/mL)	3428 ± 2194 ^a^	3729 ± 2129 ^b^	3163 ± 1683 ^a^	3326 ± 1828 ^a^	<0.01
Progesterone on hCG day (ng/mL)	1.02 ± 0.53	1.08 ± 0.53	1.07 ± 0.66	1.05 ± 0.62	0.06
LH on hCG day (IU/L)	1.64 ± 1.77	1.5 ± 2.6	1.55 ± 1.62	1.65 ± 1.7	NS

Note: In each row, values with different superscript letters indicate significant differences (*p* < 0.05). BMI, body mass index; AMH, anti-Mullerian hormone; COH, controlled ovarian hyperstimulation; GnRHa, gonadotropin-releasing hormone agonist; PPOS, progestin-primed ovarian stimulation protocol; Gn, gonadotropin; hCG, human chorionic gonadotropin; LH, luteinizing hormone; NS, not significantly different.

**Table 2 jcm-12-01993-t002:** Embryonic laboratory outcomes among four groups.

	Short-Term IVF	Partial r-ICSI	Total r-ICSI	ICSI	*p* Value
Cycles	7148	451	167	1744	
No. of oocyte retrieval	14.23 ± 7.26 ^a^	15.27 ± 6.54 ^b^	12.45 ± 5.79 ^c^	14.03 ± 6.89 ^a^	<0.001
2PN rate	62.03 ± 20.48 ^a^	57.75 ± 18.63 ^b^	65.31 ± 23.09 ^a^	72.61 ± 20.31 ^c#^	<0.001
≥3PN rate	8.94 ± 10.26 ^a^	3.15 ± 5.88 ^b^	2.19 ± 4.12 ^c^	1.50 ± 6.23 ^c^	<0.001
1PN rate	3.58 ± 6.23	3.72 ± 4.78	3.99 ± 6.79	3.26 ± 6.15	NS
No. of TQE on D3	4.58 ± 4.07 ^a^	4.03 ± 3.38 ^b^	4.50 ± 3.18 ^ab^	4.13 ± 3.57 ^b^	<0.001
Rate of TQE on D3	57.59 ± 28.34 ^a^	52.02 ± 30.25 ^b^	55.93 ± 30.63 ^ab^	52.92 ± 31.48 ^b^	<0.001
BL rate	58.63 ± 25.31 ^a^	55.18 ± 25.91 ^b^	58.85 ± 24.08 ^ab^	55.51 ± 27.72 ^b^	<0.001
Top-quality BL rate	48.51 ± 24.06 ^a^	45.1 ± 23.34 ^b^	47.47 ± 25.07 ^ab^	44.68 ± 25.79 ^b^	<0.001
No. of D5-BL	2.44 ± 2.26 ^a^	1.61 ± 1.77 ^b^	1.52 ± 1.85 ^b^	1.98 ± 2.14 ^d^	<0.001
No. of D6-BL	0.83 ± 1.34 ^a^	1.23 ± 1.31^b^	1.46 ± 1.38 ^b^	0.89 ± 1.33 ^a^	<0.001
Fresh embryo transfers, n (%)	3812 (53.33) ^a^	202 (44.79) ^ab^	78 (46.71) ^ab^	769 (44.09) ^b^	<0.001
Cleavage-stage ET	2429 (63.72)	138 (68.32)	57 (73.08)	518 (67.36)	NS
Blastocyst ET	1383 (36.28)	64 (31.68)	21 (26.92)	251 (32.64)	
Freeze-all cycle	3165 (44.28) ^a^	236 (52.33) ^b^	79 (47.31) ^ab^	897 (51.43) ^b^	<0.001
TFF cycle	19 (0.27)	0 (/)	2 (1.20)	8 (0.46)	NS
Cycle without available embryo, n (%)	152 (2.13) ^a^	13 (2.88) ^a^	8 (4.79) ^b^	70 (4.01) ^b^	<0.001

Note: In each row, values with different superscript letters indicate significant differences (*p* < 0.05). ^#^ 2PN/MII oocytes; PN, pronuclei; BL, blastocyst; TFF, total fertilization failure; ET, embryo transfer; TQE, top-quality embryos; NS, not significantly different. Rate of TQE on D3: total no. of grade 1 and 2/no. of 2PN; Top-quality BL rate: total no. of blastocyst graded as BB, BA, AB, and AA/no. of embryos for blastocyst culture; Cycle without available embryo: cycle cancellation due to no embryo available for transfer or cryopreservation.

**Table 3 jcm-12-01993-t003:** Pregnancy, delivery, and neonatal outcomes among four groups in fresh cleavage-stage ET cycles.

	Short-Term IVF	Partial r-ICSI	Total r-ICSI	ICSI	*p*-Value
Fresh cleavage-stage ETs	2429	138	57	518	
No. of embryos transferred	1.84 ± 0.37	1.86 ± 0.33	1.89 ± 0.36	1.86 ± 0.33	NS
Clinical pregnancy rate (% per ET)	51.96 (1262/2429)	55.80 (77/138)	64.91 (37/57)	50.58 (262/518)	NS
Implantation rate (% per ET embryos)	34.08 (1640/4812)	35.53 (97/273)	43.36 (49/113)	32.20 (332/1031)	NS
Pregnancy loss rate (% per CP)	14.42 (182/1262)	6.49 (5/77)	8.16 (4/37)	16.41 (43/262)	NS
Ectopic pregnancy rate (% per CP)	2.06 (26/1262)	1.30 (1/77)	3.51 (2/37)	1.53 (4/262)	NS
Live birth rate (% per ET)	43.31 (1052/2429)	51.45 (71/138)	54.39 (31/57)	41.51 (215/518)	NS
Preterm birth rate (% per LB)	21.67 (228/1052)	18.31 (13/71)	22.58 (7/31)	18.14 (39/215)	NS
C-section rate (% per LB)	78.99 (831/1052)	67.61 (48/71)	77.42 (24/31)	75.81 (163/215)	NS
Twin birth rate (% per LB)	30.99 (326/1052)	25.35 (18/71)	32.26 (10/31)	22.79 (49/215)	NS
Sex ratio (Male: female)	1.15 ^a^ (738:640)	0.89 ^b^ (42:47)	0.41 ^b^ (12:29)	0.83 ^b^ (120:144)	0.002
Mean birth weight, g	3043 ± 716	3025 ± 685	3051 ± 692	3029 ± 722	NS
LBW (% per neonate)	4.21 (58/1378)	4.49 (4/89)	4.88 (2/41)	4.55 (12/264)	NS
VLBW (% per neonate)	0.58 (8/1378)	/ (0/89)	0 (0/41)	0.758 (2/264)	NS

Note: In each row, values with different superscript letters indicated significant differences (*p* < 0.05). CP, clinical pregnancy; ET, embryo transfer; LB, live birth; Preterm birth: delivery at <37 completed weeks gestation; LBW, low birth weight, weight of neonate 1500–2499 g at birth; VLBW, very low birth weight, weight of neonate <1500 g at birth.

**Table 4 jcm-12-01993-t004:** Pregnancy, delivery, and neonatal outcomes among four groups in fresh blastocyst ET cycles.

	Short-Term IVF	Partial r-ICSI	Total r-ICSI	ICSI	*p*-Value
Blastocyst ETs	1383	64	21	251	
No. of embryos transferred	1.33 ± 0.47	1.41 ± 0.48	1.38 ± 0.48	1.35 ± 0.42	NS
Clinical pregnancy rate (% per ET)	59.22 ^a^ (819/1383)	26.56 ^b^ (17/64)	33.33 ^b^ (7/21)	58.96 ^a^ (148/251)	<0.001
Implantation rate (% per ET)	45.08 ^a^ (929/2061)	21.11 ^b^ (19/90)	31.25 ^b^ (10/32)	49.47 ^a^ (187/378)	<0.001
Pregnancy loss rate (% per CP)	8.67 (71/819)	11.76 (2/17)	28.57 (2/7)	9.46 (14/148)	NS
Ectopic pregnancy rate (% per CP)	2.08 (17/819)	/ (0/17)	/ (0/7)	2.03 (3/148)	NS
Live birth rate (% per ET)	50.69 ^a^ (701/1383)	23.43 ^b^ (15/64)	23.81 ^b^ (5/21)	51.79 ^a^ (130/251)	<0.001
Preterm birth rate (% per LB)	8.13 ^a^ (57/701)	/ ^ab^ (0/15)	20.00 ^ab^ (1/5)	14.62 ^b^ (19/130)	0.05
C-section rate (% per LB)	75.61 (530/701)	80.00 (12/15)	80.00 (4/5)	75.38 (98/130)	NS
Twin birth rate (% per LB)	18.54 (130/701)	6.67 (1/15)	20.00 (1/5)	26.92 (35/130)	NS
Sex ratio (Male:female)	1.16 (447:384)	1.29 (9:7)	/ (6:0)	1.04 (84:81)	NS
Mean birth weight, g	3128 ± 746	3105 ± 728	3119 ± 732	3095 ± 743	NS
LBW (% per neonates)	4.09 (34/831)	6.25 (1/16)	0 (0/6)	6.06 (10/165)	NS
VLBW (% per neonates)	0.48 (4/831)	/ (0/16)	0 (0/6)	0.61 (1/165)	NS

Note: In each row, values with different superscript letters indicate significant differences (*p* < 0.05). CP, clinical pregnancy; ET, embryo transfer; LB, live birth; Preterm birth: delivery at <37 completed week’s gestation; LBW, low birth weight, weight of neonate 1500–2499 g at birth; VLBW, very low birth weight, weight of neonate < 1500 g at birth.

**Table 5 jcm-12-01993-t005:** Pregnancy, delivery, and neonatal outcomes from r-ICSI derived frozen-thawed ET cycles.

	Partial r-ICSI		Total r-ICSI		*p*-Value
	Cleavage-Stage	Blastocyst	Cleavage-Stage	Blastocyst	
Total cycles of FET with transfers	142	242	62	126	
Age at transfer (years)	32.22 ± 4.16	31.14 ±4.08	32.26 ± 4.34	31.75 ± 4.14	NS
Number of FET transfers, n (%)					NS
1st cycle of FET	103 (72.54)	200 (82.64)	46 (74.19)	108 (85.71)	
2nd cycle of FET	30 (21.13)	35 (14.46)	14 (22.58)	16 (12.70)	
3rd cycle of FET	9 (6.34)	6 (2.48)	2 (3.23)	2 (1.59)	
4th cycle of FET	0 (/)	1 (0.41)	0 (/)	0 (/)	
Endometrial preparation, n (%)					
Natural cycle	37 (26.06)	60 (24.79)	14 (22.58)	35 (27.78)	
HRT cycle	64 (45.07)	103 (42.56)	28 (45.16)	48 (38.10)	
GnRHa + HRT cycle	41 (28.87)	79 (32.64)	18 (29.03)	43 (34.13)	
No. of embryos transferred	1.88 ± 0.39 ^a^	1.32 ± 0.45 ^b^	1.87 ± 0.38 ^a^	1.36 ± 0.44 ^b^	<0.001
Implantation rate (% per ET embryos)	30.58 ^a^ (85/278)	49.42 ^b^ (170/344)	32.77 ^a^ (39/119)	51.12 ^b^ (91/178)	<0.001
Clinical pregnancy rate (% per ET)	50.00 (71/142)	57.02 (138/242)	53.23 (33/62)	58.73 (74/126)	NS
Pregnancy loss rate (% per CP)	4.22 (3/71)	10.14 (14/138)	6.06 (2/33)	9.46 (7/74)	NS
Ectopic pregnancy rate (% per CP)	1.41 (1/71)	2.17 (3/138)	3.03 (1/33)	1.35 (1/74)	NS
Live birth rate (% per ET)	46.48 (66/142)	50.00 (121/242)	48.39 (30/62)	52.38 (66/126)	NS
Preterm birth rate (% per LB)	7.58 (5/66)	9.09 (11/121)	10.00 (3/30)	10.61 (7/66)	NS
C-section rate (% per LB)	77.27 (51/66)	67.77 (82/121)	73.33 (22/30)	75.76 (50/66)	NS
Twin birth rate (% per LB)	21.21 (14/66)	23.14 (28/121)	23.33 (7/30)	25.76 (17/66)	NS
Sex ratio (Male:female)	0.86 (37:43)	0.99 (74:75)	0.94 (18:19)	1.05 (42:40)	NS
Mean birth weight, g	3089 ± 695	3125 ± 713	3105 ± 709	3133 ± 724	NS
LBW (% per neonates)	3.75 (3/80)	4.70 (7/149)	2.70 (7/37)	3.66 (3/82)	NS
VLBW (% per neonates)	1.25 (1/80)	0.67 (1/149)	/ (0/37)	/ (0/82)	NS

Note: In each row, values with different superscript letters indicate significant differences (*p* < 0.05). HRT, hormone replacement treatment; GnRHa, gonadotropin-releasing hormone agonist; CP, clinical pregnancy; ET, embryo transfer; LB, live birth; Preterm birth: delivery at <37 completed weeks gestation; LBW, low birth weight, weight of neonate 1500–2499 g at birth; VLBW, very low birth weight, weight of neonate < 1500 g at birth.

## Data Availability

The data presented in this study are available on request from the corresponding author. The data are not publicly available due to privacy and ethical restriction.

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
