# Peer review of "Cycle Characteristics and Pregnancy Outcomes of Early Rescue Intracytoplasmic Sperm Injection Cycles in Normal and Hyper-Ovarian Response Women: A Six-Year Retrospective Study"

_jcm, 2023, doi:10.3390/jcm12051993_

Round 1

Reviewer 1 Report

I recommend expressing the data also using graphs and not just tables

Author Response

Point 1: I recommend expressing the data also using graphs and not just tables.

Response 1: We appreciate the reviewer for the positive comments and insightful suggestion. In our revised manuscript, we used the graphs to better illustrate the results of live birth rates among the groups in fresh and frozen embryo transfers.  

Reviewer 2 Report

The word" non-poor ovarian response"----to be replaced by normal response and or hyper response if hyper response is included

Title also need to be changed as per the above

The flow chart shows 939--poor response excluded --- how is this poor response categorised?

"the purpose of the study was to analyse the cyclic charecteristics between short term ivf, r ICSI, and iCSI cycles from normal ovarian response women in their first ivf attempts, risk factors for fertilisation failures"

The results section shows AMH in the normal range in 3 groups --Is it mean AMH or median AMH

Table No 4 shows the pregnancy rate to be low in partial  r ICSI (26.56) and the numbers in r -total ICSI are few ---explaination regarding comparision of the pregnancy rate among the groups.

Conclusions are not inline with the objectives... to be reworded and reframed.

Author Response

Point 1: The word" non-poor ovarian response"----to be replaced by normal response and or hyper response if hyper response is included.

Response 1: In this study, we included the first IVF/ICSI attempts from both normal and hyper response women. We have searched for the literature and guidelines, and currently there is no unified diagnosis criteria for hyper ovarian response, including oocyte retrieved number > 20, estradiol level > 4000 pg/mL; or oocyte retrieved number > 15, estradiol level > 5000 pg/mL (Steward RG, et al. Fertil Steril. 2014; Polat M, et al. Semin Reprod Med. 2014). Considering the controversies in strictly distinguishing normal and hyper response women in clinical practicing, we did not analyze the data for normal and hyper response respectively. We have revised this definition in our revised manuscript and we also have added this limitation in our discussion. 

Point 2: Title also need to be changed as per the above.

Response 2: Corrected.

Point 3: The flow chart shows 939--poor response excluded --- how is this poor response categorised?

Response 3: In this study, the poor ovarian response patients were defined by AMH < 1.1 ng/mL, number of oocyte retrieval < 5, and age > 42. Considering the inferior pregnancy outcomes and potential risk of fertilization failure in oocytes from decreased ovarian reserve and/or poor ovarian response due to zona pellucida thickening or poor oocyte quality, these patients were also excluded in this study. Similarly, the diagnosis criteria of poor ovarian response is varied in literature and guidelines, and we acknowledge that this exclusion criteria is less rigorous compared to Poseidon criteria for the different types of POR women, and we have added this information and limitation in our revised manuscript. 

Point 4: "the purpose of the study was to analyse the cyclic charecteristics between short term ivf, r ICSI, and iCSI cycles from normal ovarian response women in their first ivf attempts, risk factors for fertilisation failures".

Response 4:  Thank you for pointing out our inaccurate expression, we have modified this sentence in our revised manuscript.   

“The purpose of this study was to analyze the cyclic characteristics between short-term IVF, r-ICSI, and ICSI cycles from normal and hyper ovarian response women in their first IVF attempts, and identify the potential risk factors of fertilization failure.”

Point 5: The results section shows AMH in the normal range in 3 groups --Is it mean AMH or median AMH?

Response 5: Thank you for this question. The AMH level was presented as mean ± SD, and we have added this explanation for data presentation in Statistical analysis in our revised manuscript.

Point 6: Table No 4 shows the pregnancy rate to be low in partial  r ICSI (26.56) and the numbers in r -total ICSI are few ---explaination regarding comparison of the pregnancy rate among the groups.

Response 6: We appreciate for this insightful comment and suggestion. We totally agree with you that the pregnancy rate was low in r-ICSI groups regarding to blastocyst transfers, and the small sample size in r-ICSI groups in this fresh blastocyst transfer comparison. The possible reasons for this inferior pregnancy outcomes could be delayed blastocyst development and the time shift in fecundation. The blastocyst formation rate and top-quality blastocyst rate were decreased in r-ICSI groups compared to IVF group, indicating that delayed fertilization by 5-6 h may postpone embryo development at the blastocyst stage and possibly lead to asynchronization between blastocyst and endometrial receptivity. However, the samples size for fresh blastocyst transfers in r-ICSI groups was small, the reasons for dissatisfied pregnancy outcomes in fresh blastocyst transfer and the underlying mechanisms need to be further verified. Given the low frequency of r-ICSI in IVF cycles, and the fresh embryo transfers were performed on day 3, therefore the sample size of fresh blastocyst transfers was small, the reasons for dissatisfied pregnancy outcomes in fresh blastocyst transfer and the underlying mechanisms need to be further verified. We have added the explanation in Discussion in our revised manuscript. 

Point 7: Conclusions are not in line with the objectives... to be reworded and reframed.

Response 7: We appreciate this comment and suggestion, we have revised our conclusions to in line with the objectives.  

Reviewer 3 Report

In the present manuscript entitled Cycle characteristics and pregnancy outcomes of early rescue intracytoplasmic sperm injection cycles in non-poor ovarian response women: a six-year retrospective study, the authors described cyclic characteristics, pregnancy, delivery and neonatal outcomes between short-term IVF, r-ICSI, and ICSI cycles from normal ovarian response women in their first IVF attempts.

The conducted research is interesting and well-written and limitations of the study have been taken into account. However, below there are minor points, which should be taken into account.

1. In all figures and tables an explanation of all abbreviations used should be included in the caption (e.g. Figure 1 – TFF, IVF, ICSI; Table 1 – BMI, AMH, PPOS)

2. In the sentence: Briefly, the MII oocyte was immobilized by holding (Sunlight,) with the first PB at 12 o’clock, and the immobilized sperm was microinjected into the cytoplasm at 3 o’clock. (line 137, 138) the country should be placed after the manufacturer's name "Sunlight".

3. The authors should provide the explanation of all abbreviations in the main text of manuscript, where they were first used (e.g. E2 – line 209, AMH – line 86)

Author Response

In the present manuscript entitled Cycle characteristics and pregnancy outcomes of early rescue intracytoplasmic sperm injection cycles in non-poor ovarian response women: a six-year retrospective study, the authors described cyclic characteristics, pregnancy, delivery and neonatal outcomes between short-term IVF, r-ICSI, and ICSI cycles from normal ovarian response women in their first IVF attempts.

The conducted research is interesting and well-written and limitations of the study have been taken into account. However, below there are minor points, which should be taken into account.

We appreciate the reviewer for the insightful comments, suggestions and positive responses to our study. Please find below our point-to-point responses to the comments.

Point 1: In all figures and tables an explanation of all abbreviations used should be included in the caption (e.g. Figure 1 – TFF, IVF, ICSI; Table 1 – BMI, AMH, PPOS).

Response 1: Thank you for this suggestion, the explanation of all abbreviations were added in our revised manuscript.

Point 2: In the sentence: Briefly, the MII oocyte was immobilized by holding (Sunlight,) with the first PB at 12 o’clock, and the immobilized sperm was microinjected into the cytoplasm at 3 o’clock. (line 137, 138) the country should be placed after the manufacturer's name "Sunlight".

Response 1: Thank you for pointing our mistakes. These information were added in our revised manuscript.

Point 3: The authors should provide the explanation of all abbreviations in the main text of manuscript, where they were first used (e.g. E2 – line 209, AMH – line 86).

Response 3: Thank you for your suggestion, we have carefully examined our manuscript and the explanation were supplemented for all abbreviations.  

Reviewer 4 Report

This is an interesting article about the use of early rescue intracytoplasmic sperm injection (r-ICSI) in the first cycles in non-poor ovarian response women. The authors demonstrate that early r-ICSI is an efficacy and safe option for the embryologist in proceeding with convention IVF in couples without male infertility. However, some issues could be elaborated on: - r-ICSI is performed after the second assessment of PB extrusion, 4-6 hours after IVF according to the authors. It means that this procedure could be only possible in laboratories working "full time" but also in centres performing oocytes pick up only in the morning, otherwise the r-ICSI procedures would take place at night or in the evening. - it is clear from the text that it would be better to perform a FET in embryos obtained after r-ICSI, due to the time shift in the fecondation. However, the datas presented for the FET cycles (for example patient information, number of transfers performed per patient) are far less than the datas provided for the fresh cycles. 

Author Response

Point 1: This is an interesting article about the use of early rescue intracytoplasmic sperm injection (r-ICSI) in the first cycles in non-poor ovarian response women. The authors demonstrate that early r-ICSI is an efficacy and safe option for the embryologist in proceeding with convention IVF in couples without male infertility. However, some issues could be elaborated on: - r-ICSI is performed after the second assessment of PB extrusion, 4-6 hours after IVF according to the authors. It means that this procedure could be only possible in laboratories working "full time" but also in centres performing oocytes pick up only in the morning, otherwise the r-ICSI procedures would take place at night or in the evening. - it is clear from the text that it would be better to perform a FET in embryos obtained after r-ICSI, due to the time shift in the fecondation. However, the datas presented for the FET cycles (for example patient information, number of transfers performed per patient) are far less than the datas provided for the fresh cycles. 

Response 1: We appreciate the reviewer for the positive comments and insightful suggestions. Traditionally in our center, all the procedures involved in controlled ovarian stimulation and embryo laboratory, including hormone detection, follicle measurement, Gn injection, hCG trigger, oocyte retrieval, in vitro insemination, embryo transfer, and embryo cryopreservation and thawed, are performed in a relatively fixed schedule time. In specific, all the hormone level detection and follicle measurements will be finished before 12:00, the hCG trigger will be injected between 08:00-11:00 pm, and oocyte retrieval will be performed between 08:00-11:00 am based on the number of patients. According to our knowledge, these are the standard operating procedures in most of the reproductive centers in China, although early r-ICSI is an alternative option for late r-ICSI or extended application of ICSI. Regarding to the early r-ICSI procedure in our center, one senior embryologist will be assigned and informed to take night-shift for performing r-ICSI at 17:00-19:00. We have added the time schedule explanation for hCG triggering, oocyte retrieval, and in vitro insemination in our revised manuscript. We understand that early r-ICSI may potentially increase the workflow burden for embryo laboratories, while the excessive use of ICSI, aiming at increasing fertilization rate, does not improve the success rate in couples without male infertility, but with higher costs and also increased laboratory workload. We hope our study and results can provide some referential significance to illustrate the beneficial role of early r-ICSI, and to increase the confidence of embryologists in performing conventional IVF in couples without male infertility.

We thank the reviewer for the suggestions regarding to FET cycles. In our revised manuscript, more data were supplemented for FET cycles, including the endometrium preparation in Material and Methods, and more patient information were added in Table 5.

Round 2

Reviewer 4 Report

the authors have argued the reviewer's comments and the work has been improved and can stimulate discussion in the scientific field